# *Post*: Device Placement with Cross-Entropy Minimization and Proximal Policy Optimization

**Yuanxiang Gao**[1,2]      **Li Chen** [3]      **Baochun Li** [1]

[1] Department of Electrical and Computer Engineering, University of Toronto
[2] School of Information and Communication Engineering,
University of Electronic Science and Technology of China
[3] School of Computing and Informatics, University of Louisiana at Lafayette
yuanxiang@ece.utoronto.ca,  li.chen@louisiana.edu,  bli@ece.toronto.edu

## Abstract

Training deep neural networks requires an exorbitant amount of computation resources, including a heterogeneous mix of GPU and CPU devices. It is critical to place operations in a neural network on these devices in an optimal way, so that the training process can complete within the shortest amount of time. The state-of-the-art uses reinforcement learning to learn placement skills by repeatedly performing Monte-Carlo experiments. However, due to its equal treatment of placement samples, we argue that there remains ample room for significant improvements. In this paper, we propose a new joint learning algorithm, called *Post*, that integrates cross-entropy minimization and proximal policy optimization to achieve theoretically guaranteed optimal efficiency. In order to incorporate the cross-entropy method as a sampling technique, we propose to represent placements using discrete probability distributions, which allows us to estimate an optimal probability mass by maximal likelihood estimation, a powerful tool with the best possible efficiency. We have implemented *Post* in the Google Cloud platform, and our extensive experiments with several popular neural network training benchmarks have demonstrated clear evidence of superior performance: with the same amount of learning time, it leads to placements that have training times up to 63.7% shorter over the state-of-the-art.

## 1   Introduction

With an increasing demand of computing resources to train today's deep neural networks (DNNs), it becomes typical to leverage a heterogeneous mix of both CPU and GPU devices [1, 2]. In such a distributed training environment, it is important to specify how each operation in a neural network should be placed on each of these CPU and GPU devices, referred to as the *device placement* problem. The objective is to find a *placement* of operations on devices, so that the time required to train the neural network can be minimized.

In the recent literature, Mirhoseini *et al.* [3, 4] proposed to solve the device placement problem with a reinforcement learning approach, based on the policy gradient method [5]. Unfortunately, this method is inefficient because it relies on the Monte Carlo method to generate samples, which treats each data sample equally without emphasizing important ones. To improve the learning efficiency, an importance sampling technique, called the cross-entropy method [6, 7, 8], becomes promising, due to its theoretical guarantee on the optimal efficiency when generating samples [9].

In this paper, our first contribution is to apply the importance sampling technique in device placement and model the problem as a cross-entropy minimization problem. An example of the placement probability distribution for a DNN with two operations is shown in Fig. 1. Each possible placement is

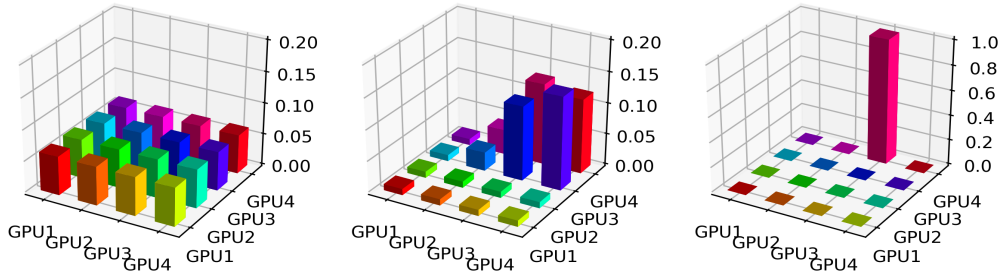

Figure 1: Adjusting the probability distribution of the placement for two operations with cross-entropy minimization. The x-axis represents the device to place the first operation, the y-axis represents the device to place the second operation, and the z-axis represents the probability of selecting a placement.

associated with a probability, according to a two-dimensional probability mass function. Initially, all the placements have an equal probability (left). With cross-entropy minimization, a better probability distribution can be learned, where the placement with a shorter training time has a higher probability (middle). Eventually, the probability distribution will converge to a degenerate distribution, with which the optimal placement has a probability of 1.

As cross-entropy minimization adjusts the distribution after a batch of placement trials, it lacks a reinforcement learning algorithm with a finer granularity, one which has the ability to fine-tune the distribution after every placement trial. Our original contribution in this paper focuses on the design and implementation of a new learning algorithm, called *Post*, that integrates cross-entropy minimization and proximal policy optimization [10, 11], a state-of-the-art reinforcement learning algorithm. With *Post*, cross-entropy minimization is applied to each batch of trials, to reduce the sample space of placement for a higher sample efficiency. Within a batch, proximal policy optimization is applied to each placement trial to incrementally improve the placement distribution, which achieves a further performance boost in the learning process.

We have evaluated *Post* with a wide variety of deep neural network models in TensorFlow, including Inception-V3 [12], ResNet [13], Language Modeling [14] and Neural Machine Translation [15]. For an extensive array of experiments, we have deployed *Post* on up to 96 CPU and GPU nodes in the Google Cloud platform for distributed training. Our experimental results have demonstrated a substantial improvement in the learning efficiency achieved by *Post*, compared with using the policy gradient method [3], proximal policy optimization only [16], and cross-entropy minimization only. Thanks to its efficiency, *Post* is able to find better placements for ResNet, Inception-V3 and NMT benchmarks, respectively, that lead to 24.0% to 63.7% shorter training time than the policy gradient method [3]. Further, detailed runtime profiles have clearly shown that *Post* is able to discover a non-trivial placement "trick" to speedup the training process: it is able to parallelize the model without incurring undesirable communication delays across GPUs.

## 2 Related Work

We classify the existing research efforts on device placement for deep neural networks into three categories as follows.

**Device placement with reinforcement learning.** Mirhoseini, *et al.* [3] proposed a sequence-to-sequence RNN model as the parameterized policy for generating placements. As RNN suffers from the vanishing (and exploding) gradient problem when predicting long sequences, this work manually assigns operations into hundreds of groups and further places groups on devices, thus reducing the length of the input sequence. More recently, they replaced the manual grouping with an automated grouping mechanism [4], which is a feedforward neural network. Both works used the policy gradient method to train their RNNs; in contrast, Spotlight [16] applied proximal policy optimization to achieve a faster training speed. This work adopts a different approach, using the softmax distributions rather than an RNN as the policy representation. Accompanied by proximal policy optimization and

cross-entropy minimization, training softmax distributions solves the device placement problem with a lower training overhead, as demonstrated in our extensive experiments.

**Cross-entropy method.** Initially developed for variance reduction in rare-event simulations [9], the cross-entropy method has been shown as an efficient method to solve combinatorial optimization problems by treating the optimal solution as a rare-event to be discovered [17]. Due to its efficiency, the cross-entropy method achieves several orders of magnitude higher scores than policy gradient method in Tetris games [8]. This work adapts the cross-entropy method for device placement, by modeling device placement as multi-dimensional softmax distributions. We have derived a closed-form solution of cross-entropy minimization for softmax distributions, which has not been developed in the existing literature. Our proposed algorithm integrates the cross-entropy method with proximal policy optimization, which further improves its efficiency.

**Proximal policy optimization.** The theory of proximal policy optimization was proposed by Shulman, *et al.* [18, 11], where an objective function was derived to obtain the performance lower bound of the new policy. This method shows superior performance in high-dimensional continuous control problems. Heess, *et al.* [10] proposed a distributed algorithm for proximal policy optimization, which can successfully teach a simulated agent to perform complex tasks. Our work demonstrates that by incorporating periodic cross-entropy minimization for aggressive policy improvement, the learning performance of proximal policy optimization can be significantly improved.

## 3 Algorithm Design

### 3.1 Device Placement: Preliminaries

For a deep neural network with $M$ operations, a particular placement is a vector of devices assigned to each of its operations, represented as $d = \{d_1, d_2, \cdots, d_m, \cdots, d_M\}$, where $d_m \in \{\text{CPU}, \text{GPU}_0, \text{GPU}_1, \cdots\}$ is the device selected from the set of available devices to run the $m$-th operation. Given a placement $d$, the time it takes to train the neural network is denoted by $T(d)$. Our goal is to find an optimal placement, so that the training time using this placement is minimized. Formally, the device placement problem can be formulated as

$$\gamma^* = \min_{d \in A} T(d), \tag{1}$$

where $A$ denotes the set of all possible placements of the deep neural network.

For such a combinatorial optimization problem, Mirhoseini, *et al.* [3, 4] generated placement samples from a placement distribution and used the policy gradient method [5] to improve parameters of the distribution, so that a better placement can be generated with a higher probability. However, the policy gradient method is sample inefficient [18], as it performs only a one-step gradient update for each data sample [11].

To achieve a higher sample efficiency, a more advanced policy gradient method — proximal policy optimization — was proposed in the recent literature of reinforcement learning [11, 10]. The main idea is to derive a performance objective and perform multiple steps of gradient updates towards optimizing the objective. Still, both policy gradient and proximal policy optimization perform only one step or several steps of stochastic gradient descent/ascent (SGD/SGA), which are slow as they do not achieve the optimum. In contrast, the unique advantage of cross-entropy minimization is that it formulates the performance improvement as an optimization problem, which is directly solved to obtain the optimal solution.

Although optimization-based cross-entropy minimization improves performance faster than SGD/SGA-based methods, it updates parameters after each batch of placement samples is generated, without online per-sample parameter updates as in SGD/SGA-based methods. Therefore, it is intuitive that an elegantly designed combination of these two methods brings the fastest and the most frequent parameter improvement.

### 3.2 Cross-Entropy Minimization for Device Placement

For the placement of the $m$-th operation, we use a parameter $u_{mj}$ to represent the preference of choosing the $j$-th device. With a softmax function, we normalize the parameters for all the devices to

obtain a probability distribution of the placement. Particularly, the probability of choosing the $j$-th device for the $m$-th operation is given by:

$$f(d_m = j|u_m) = \frac{e^{u_{mj}}}{\sum_{i=1}^{D} e^{u_{mi}}},\tag{2}$$

where $D$ is the total number of devices, and $u_m$ denotes the vector of parameters $\{u_{m1}, u_{m2}, \cdots, u_{mD}\}$ for the $m$-th operation.

Similarly, we use $u$ to denote the vector of parameters $\{u_1, u_2, \cdots, u_M\}$ for the entire neural network. Observing that the placements of all the operations in the network can be configured independently in practice, the device selection for each operation can also be independent. Thus, the joint distribution $f(d|u)$, *i.e.*, the placement distribution for the neural network, can be expressed as a product of marginal distributions for each operation:

$$f(d|u) = \prod_{m=1}^{M} f(d_m|u_m).\tag{3}$$

To obtain the optimal distribution which results in the optimal placement $d^* = \arg\min T(d)$ with probability 1, we iteratively update the parameters until the final convergence, starting from randomly generated values. To be particular, initially with parameter $u^{(0)}$, we gradually improve the joint distribution through a set of target distributions along iterations, with parameters $u^{(1)}, u^{(2)}, \cdots$. The target distribution at the $t$-th iteration is defined as the following conditional distributions:

$$f(d|u^{(t+1)}) = f(d|u^{(t)}, T(d) \leq \gamma_t),\tag{4}$$

where $\gamma_t$ is a constant and $\gamma_0 > \gamma_1 > \cdots > \gamma^*$.

According to the definition of conditional probability, Eq. (4) can be expressed as

$$f(d|u^{(t+1)}) = \frac{f(d, T(d) \leq \gamma_t|u^{(t)})}{f(T(d) \leq \gamma_t|u^{(t)})} = \frac{I_{\{T(d) \leq \gamma_t\}} f(d|u^{(t)})}{\sum_d I_{\{T(d) \leq \gamma_t\}} f(d|u^{(t)})},\tag{5}$$

where $I_{\{\cdot\}}$ is the indicator function which equals to 1 when the condition holds and to 0 otherwise. Intuitively, Eq. (5) represents a normalized density of the good placements (which result in training times smaller than $\gamma_t$) in $f(d|u^{(t)})$. With the normalization, the old distribution $f(d|u^{(t)})$ is updated to the target distribution $f(d|u^{(t+1)})$. As $\gamma_t$ decreases over iterations, the probability of generating good placements from the target distribution keeps increasing.

With such an intuition, we now describe how parameters are updated at each iteration $t$ in detail. The parameters $u^{(t)}$ will be adjusted by minimizing the stochastic distance, specifically, the Kullback-Leibler (KL)-divergence, between the old distribution and the target distribution, which can be expressed as

$$\min_{u^{(t)}} \sum_d f(d|u^{(t+1)}) \log f(d|u^{(t+1)}) - f(d|u^{(t+1)}) \log f(d|u^{(t)}),\tag{6}$$

where the target distribution $f(d|u^{(t+1)})$ is given in Eq. (5). As the first term in the objective is constant with respect to $u^{(t)}$, the KL-divergence minimization is equivalent to the following cross-entropy minimization:

$$\min_{u^{(t)}} -\sum_d I_{\{T(d) \leq \gamma_t\}} f(d|u^{(t)}) \log f(d|u^{(t)}),\tag{7}$$

which can be further transformed to its equivalent expectation form:

$$\min_{u^{(t)}} -E_{u^{(t)}}[I_{\{T(d) \leq \gamma_t\}} \log f(d|u^{(t)})].\tag{8}$$

To solve this problem, we replace the expectation with its average over $N$ samples and minimize the sample average as the following:

$$\min_{u^{(t)}} -\frac{1}{N} \sum_{n=1}^{N} I_{\{T(d^{(n)}) \leq \gamma_t\}} \log f(d^{(n)}|u^{(t)}),\tag{9}$$

where $d^{(n)}$ is the $n$-th sampled placement. Among the $N$ sampled placements, the subset satisfying the condition is determined by the constant $\gamma_t$. Using the typical cross-entropy method [17], we set $\gamma_t$ so that the percentage of placements satisfying the condition is $\rho$ (e.g., $\rho = 0.1$). To be more specific, $\gamma_t$ is set as the $\rho N$-th lowest training time among all the sampled placements. The cross-entropy minimization in Eq. (9) is a maximum likelihood estimator (MLE) for estimating the target distribution, which has theoretically guaranteed optimal sample efficiency [19]. As a convex optimization problem [20], problem (9) can be efficiently solved by a gradient-based method to achieve the global minimum.

Further, we have derived the following closed-form solution to problem (9), for softmax distributions.

**Theorem 1.** *The solution of the cross-entropy minimization problem in (9) for softmax distributions has the following closed form:*

$$f(d_m = j | u_m^{(t+1)}) = \sum_{p=1}^{P} \frac{I_{\{d_m^{(p)} = j\}}}{P}, \tag{10}$$

*where $p$ refers to the promising placements and $P = \rho N$ is the total number of promising placements.*

*Proof:* Please refer to the supplementary material for a proof of this theorem.

Eq. (10) gives an efficient and gradient-free approach to solve the cross-entropy minimization problem. Among the promising placements, we count the frequency of the $j$-th device allocated to the $m$-th operation, and use it as the placement probability given new parameters.

## 3.3 Joint Learning with Proximal Policy Optimization

Cross-entropy minimization is a batch learning algorithm as it updates parameters after a large number ($N$, which can be hundreds) of placement samples. Such a batch learning algorithm lacks the incremental parameter improvement in an online reinforcement learning for every small number ($K$) of placement samples. Therefore, we integrate cross-entropy minimization and a recent reinforcement learning algorithm, called proximal policy optimization [10, 18, 11], to achieve a further learning speedup. The objective of proximal policy optimization at the $t$-th iteration is as follows:

$$\max_{u^{(t+1)}} \frac{1}{K} \sum_{n=tK+1}^{tK+K} \sum_{m=1}^{M} \frac{f(d_m^{(n)} | u_m^{(t+1)})}{f(d_m^{(n)} | u_m^{(t)})} (b - T(d^{(n)})) - \beta D_{KL}(f(d_m^{(n)} | u_m^{(t)}) || f(d_m^{(n)} | u_m^{(t+1)})), \tag{11}$$

where $b$ is the moving average of the training times across all the sampled placements, and $D_{KL}(\cdot)$ is the KL divergence between the old and new distributions. The coefficient $\beta$ is an adaptive hyperparameter, whose value is adjusted each iteration based on the calculated KL divergence and a target value of KL divergence [11] so as to avoid policy updates that are too large or too small. The term $b - T(d^{(n)})$ represents a reward signal for a placement, which is positive if its training time is smaller than the average and negative otherwise. This objective is a performance lower bound of distributions under the new parameters [18]. Optimizing the lower bound leads to performance improvement over the old parameters.

Our joint policy optimization algorithm, *Post*, is summarized in Algorithm 1. Within a loop, *Post* continuously samples placements from the distribution and evaluates their training times. For every $K$ sampled placements, we perform several (*e.g.*, 10) stochastic gradient ascent steps with respect to the objective of proximal policy optimization, which makes incremental policy improvements. For every $N$ sampled placements ($N$ is several or tens of times larger than $K$), we solve the cross-entropy minimization problem with Eq. (10) to achieve a global and an aggressive policy improvement.

After each cross-entropy minimization, the probability of choosing a particular device for an operation may become zero, which discourages the exploration of more potential placements. For better exploration, we mix the distribution with a uniform distribution, resulting in a probability of $\epsilon$ (*e.g.*, 0.1) for a device to be uniformly selected from all the available devices, and a probability of $1 - \epsilon$ for a device to be chosen according to the solution of cross-entropy minimization. For softmax distributions, the update rule for stochastic gradient ascent with respect to the objective of proximal policy optimization can be expressed in closed forms, as derived in our supplementary material.

---

**Algorithm 1** *Post*: Joint Policy Optimization

---
1: Initialize parameters $u^{(0)}$ as all zeros; Initialize $t = 0$;
2: **for** $n = 1, 2, \ldots, L$ **do**
3:     Sample a placement $d^{(n)} \sim f(d|u^{(t)})$;
4:     Train the DNN under $d^{(n)}$ and recording $T(d^{(n)})$;
5:     **if** $n\%K == 0$ and $n\%N \neq 0$ **then**
6:         Perform several (*e.g.*, 10) stochastic gradient ascent steps w.r.t. the objective of proximal policy optimization in Eq. (11);
7:         $t = t + 1$
8:     **end if**
9:     **if** $n\%N == 0$ **then**
10:       Solve the cross-entropy minimization using Eq. (10) to achieve a global minimum;
11:       $t = t + 1$
12:       Mix the new distribution $f(d_m|u_m^{(t+1)}) = (1 - \epsilon)f(d_m|u_m^{(t+1)}) + \epsilon\frac{1}{D}, \forall m$;
13:     **end if**
14: **end for**

---

## 4 Experimental Evaluation

In this section, we have implemented and evaluated *Post* on the Google Cloud platform with an extensive set of deep learning models. Experimental results have demonstrated the superior performance of *Post* in comparison with existing mechanisms in the literature. A more in-depth analysis on runtime profiles has also been presented to better understand the advantages of our algorithm.

**Setup.** We have conducted our experiments with 12 machines on the Google Cloud platform. Each machine is equipped with 26 GB of main memory, an Intel Broadwell 8-core CPU and 2, 4 or 8 NVIDIA Tesla K80 GPUs, each with 11 GB of memory.

The training of *Post* progresses in a distributed fashion, with a parameter server maintaining the parameters of softmax placement distributions and 12 workers. In particular, at each iteration of learning, the parameter server samples the distributions to obtain 12 placements, each to be evaluated at a worker. After receiving the training times from all the workers, the parameter server applies proximal policy optimization to update the parameters. After five iterations, the parameter server further updates its parameters by solving cross-entropy minimization.

In our experiments, we train the placement parameters for a neural network with a total of 2400 sampled placements. For each sampled placement, a worker configures the placement accordingly and trains the network for 11 steps. As the first step involves initial configuration, we use the average training time of the following 10 steps to measure the performance of this placement.

To handle the placements that result in Out-Of-Memory (OOM), each of such placements is assigned with a large per-step training time (*e.g.*, 100 seconds). Our detailed setting of the parameters in Algorithm 1 is as follows: the learning rate of SGA is 1; the ratio for choosing promising placements is 0.1 (6 best placements out of 60 samples over 5 iterations); the exploration factor $\epsilon$ is 0.1, which is linearly reduced to zero during learning; the KL penalty coefficient $\beta$ is initialized as 1 and adapted based on a target KL divergence of 0.03.

We have evaluated *Post* using four open-source benchmarks in TensorFlow: *Inception-V3* [12, 21], *ResNet* [13, 21], *RNN Language Modeling* (RNNLM) [14], and *Neural Machine Translation (NMT) [15, 22]*. Inception-V3 and ResNet are two popular deep convolutional neural networks trained on the ImageNet dataset using a batch size of 32. RNNLM is a 4-layer RNN trained with a batch size of 64. NMT is a sequence-to-sequence encoder-decoder architecture trained on the WMT16 dataset with a batch size of 64.

We compare the training time performance — including both the average and the minimum training time — of the placements found by *Post* with the following baselines: *Single GPU,* which assigns all the operations to a single GPU, except for those without GPU implementation; *Policy Gradient Method,* the device placement algorithm proposed by Google [3, 4], with the policy gradient update rule; *Proximal Policy Optimization,* the reinforcement learning algorithm [10, 11] used by Spotlight [16]), performing SGA updates; *Cross-Entropy Minimization,* our proposed global optimization

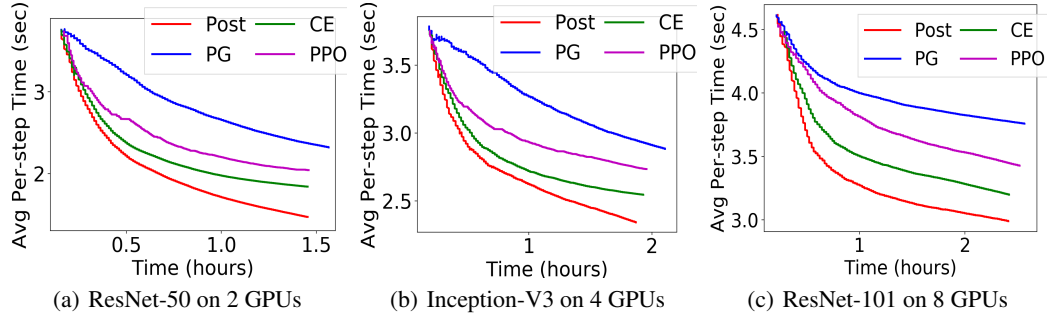

(a) ResNet-50 on 2 GPUs     (b) Inception-V3 on 4 GPUs     (c) ResNet-101 on 8 GPUs

Figure 2: Average performance of *Post*, compared with the baselines of CE — Cross-Entropy Minimization, PG — Policy gradient method, and PPO — Proximal Policy Optimization.

Table 1: Per-step training time (in seconds) of placements given by the baselines. The training time reduction is calculated based on the policy gradient method. OOM stands for Out-Of-Memory. Errors of per-step times are within 2% around the average.

| Models | GPUs | Single GPU | Expert | Metis | Policy Gradient | *Post* | Reduction |
|---|---|---|---|---|---|---|---|
| RNNLM | 2 | 0.44 | 0.89 | 1.81 | 0.66 | 0.44 | 33.3% |
| ResNet-50 | 2 | 0.86 | 0.86 | 4.47 | 1.49 | 0.86 | 42.3% |
| ResNet-200 | 2 | 2.75 | 2.75 | 11.63 | 3.62 | 2.75 | 24.0% |
| ResNet-50 | 4 | 0.86 | 0.86 | 4.21 | 2.37 | 0.86 | 63.7% |
| ResNet-101 | 8 | 1.59 | 1.59 | 7.31 | 2.90 | 1.59 | 45.2% |
| Inception-V3 | 2 | 1.79 | 1.79 | 6.76 | 2.15 | 1.30 | 39.5% |
| Inception-V3 | 4 | 1.75 | 1.75 | 6.07 | 1.82 | 1.19 | 34.6% |
| Inception-V3 | 8 | 1.78 | 1.78 | 7.35 | 2.67 | 1.29 | 51.7% |
| NMT (2-layer) | 2 | OOM | 2.05 | 7.47 | 2.30 | 1.22 | 47.0% |
| NMT (4-layer) | 4 | OOM | 3.43 | 11.04 | 4.08 | 2.13 | 47.8% |
| NMT (8-layer) | 8 | OOM | 5.11 | 12.74 | 5.35 | 2.20 | 58.9% |

method in Theorem 1, without the updates from proximal policy optimization; *Metis,* an algorithm that partitions the network into 2 parts (2-GPU case), 4 parts (4-GPU case) or 8 parts (8-GPU case), each assigned to one device, according to a cost model of operations in the network [23]; *Expert,* a method that places the entire model on a single GPU for Inception-V3 and ResNet [12, 13], and assigns each LSTM layer to a separate GPU for NMT and RNNLM, collocating the embedding layer with the first LSTM layer, the softmax layer and the attentional layer with the last LSTM layer [15].

Due to different hardware (CPU and GPU) and TensorFlow benchmark versions, a complete reproduction of the experiments in the previous works [4, 3] is not feasible. As a compromise, we have implemented and run the policy gradient method proposed in these papers in our hardware and software environments, so that a fair comparison with our proposed algorithm can be made.

**Average Performance.** Figure 2 presents the average per-step training time resulted from sampled placements for a neural network, along with the progress of learning the softmax distributions. Three neural networks are to be placed on different numbers of GPUs, corresponding to Figure 2(a)-(c), respectively.

As shown in Figure 2(a) where ResNet-50 is to be placed on 2 GPUs, the policy gradient method suffers from the slowest progress, with which the training times are always longer than 2 seconds. Proximal policy optimization improves the training time performance at a faster speed than the policy gradient method, while the cross-entropy minimization is even faster. Compared to these baselines, *Post* achieves the fastest improvement and obtains the shortest training time, as it integrates cross-entropy minimization into proximal policy optimization to simultaneously improve the placement distribution. In a similar vein, *Post* significantly outperforms the baselines when placing deeper networks on more GPUs, as demonstrated by the learning curves in Figure 2(b) and (c).

**Best Performance.** As shown in Table 1, for RNNLM and ResNet, *Post* is identical to the Single GPU baseline. This is because the architecture of these networks is not suitable to be processed in

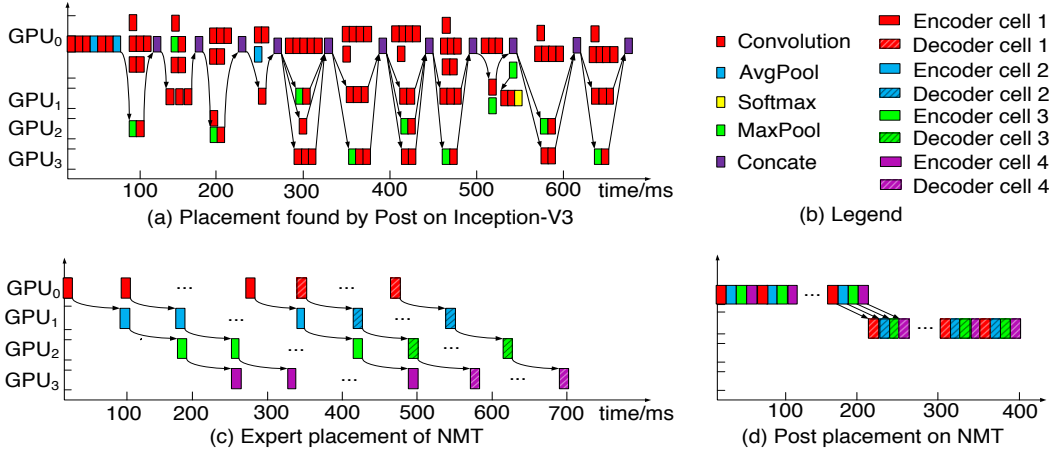

Figure 3: Performance profile of the placement found by *Post* on Inception-V3 and NMT.

parallel. For example, the continuous convolution layers of ResNet are densely connected with each other. As a result, partitioning them to multiple GPUs introduces a long data communication delay between GPUs. Due to its slower learning speed, the best placement found by the policy gradient method given the amount of learning time is far from the optimum, and is worse than the Single GPU case.

For Inception-V3, *Post* is able to find optimal placements that outperform the Single GPU case. For all three cases, the policy gradient method fails to discover a placement that is better than *Post* or Single GPU within a pre-specified amount of learning time. Specifically, *Post* outperforms the policy gradient method by a training time reduction of $39.5\%$, $34.6\%$ and $51.7\%$ for 2 GPUs, 4 GPUs and 8 GPUs, respectively. For NMT, the Single GPU baseline is no longer feasible due to the Out-Of-Memory error, and the Expert method performs the best among all the baselines. *Post* is able to find better placements that outperform the Expert method, while the policy gradient method cannot. Compared with the policy gradient method, *Post* finds better placements that reduce the training times by $47.0\%$, $47.8\%$ and $58.9\%$ for 2 GPUs, 4 GPUs and 8 GPUs, respectively.

We further present end-to-end training time reduction resulted from optimized placements discovered by *Post*. With Inception-V3 trained on 4 GPUs for 40000 steps, it takes 13.5 hours to complete the training given the placement found by *Post*. In comparison, it requires 20 hours for the single GPU case and 20.5 hours with the placement found by the policy gradient method, which clearly demonstrate that *Post* saves the end-to-end training time by $32.5\%$ and $34.1\%$, respectively.

**Analysis of Discovered Placement.** We present the placements our algorithm has discovered for Inception-V3 and NMT in Figure 3(a) and 3(d), in comparison with the placement given by the Expert method, shown in Figure 3(c).

As observed in Figure 3(a), the per-step training for Inception-V3 involves consecutive inception blocks, each with multiple parallel branches of convolutional layers. For the first four inception blocks, despite four parallel branches, *Post* only allocates one of them to a different GPU. Since the first four branches are relatively light-loaded, we believe the reason of such an allocation is that the communication overhead will outweigh the benefit of balanced load if the four branches are allocated to separate GPUs. It is interesting to observe that for the four blocks with heavier loads in the middle, *Post* increases the degree of parallelism by allocating more operations to more GPUs. With this allocation, the computation loads on $GPU_0$ and communication overhead among GPUs is well balanced so that the training time is minimized.

Figure 3(c) illustrates the placement of a sequence-to-sequence RNN, which involves recurrent encoding-decoding steps on consecutive LSTM cells. With the placement from the Expert method, each layer is placed on a separate GPU. Due to massive connections between consecutive layers, outputs of the previous layer need to be transmitted across GPUs at every step of state transition, which incurs expensive communication overhead. In comparison, *Post* discovers a novel strategy to parallelize such a sequence-to-sequence model. As shown in Figure 3(d), rather than partitioning LSTM layers into GPUs, it divides functions (encoding or decoding) into GPUs. In this way, one

GPU is dedicated for encoding tasks and another one is dedicated for decoding tasks. Such a trick learned by *Post* makes great sense, as cutting along the encoder-decoder boundaries incurs a negligible communication overhead, which greatly reduces the communication delay and accelerates the training.

## 5 Conclusion

In this paper, we represent device placement as a high-dimensional softmax probability distribution, which translates the problem of finding the best placement to one of estimating the optimal density. With such a new model, we developed a customized cross-entropy method to estimate the optimal placement, which theoretically guarantees the best possible efficiency. A highlight of our contribution is our integration of proximal policy optimization, an online reinforcement learning algorithm, and the cross-entropy method, a batch learning algorithm, to achieve a faster policy improvement. We evaluated our proposed algorithm with an array of deep neural network models in TensorFlow. Our experiments demonstrated that *Post* achieved a significantly better learning performance than the policy gradient method, proximal policy optimization, and the cross-entropy method alone.

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
