[Supplementary Material]

# *Post*: Device Placement with Cross-Entropy Minimization and Proximal Policy Optimization (Appendix)

**Yuanxiang Gao**[1,2]    **Li Chen**[3]    **Baochun Li**[1]
[1] Department of Electrical and Computer Engineering, University of Toronto
[2] School of Information and Communication Engineering,
University of Electronic Science and Technology of China
[3] School of Computing and Informatics, University of Louisiana at Lafayette
yuanxiang@ece.utoronto.ca,  li.chen@louisiana.edu,  bli@ece.toronto.edu

## 1 Proof of Theorem 1

**Theorem 1.** *The solution of the cross-entropy minimization in Eq. (9) for softmax distributions has the following closed form:*

$$f(d_m = j|u_m^{(t+1)}) = \sum_{p=1}^{P} \frac{I_{\{d_m^{(p)}=j\}}}{P}. \tag{1}$$

*where $p$ refers to the promising placements and $P = \rho N$ is the total number of promising placements.*

*Proof.* Our goal is to optimally set each parameter of softmax distributions so that the objective in Eq. (9) is minimized. For a particular parameter $u_{mj}$, the gradient of objective in Eq. (9) with respect to $u_{mj}$ can be written as,

$$-\frac{1}{N} \sum_{n=1}^{N} I_{\{T(d^{(n)}) \le \gamma_t\}} \bigtriangledown_{u_{mj}} \log \prod_{k=1}^{M} f(d_k^{(n)}|u_k^{(t)}). \tag{2}$$

For those placements with training times larger than $\gamma_t$, the indicator function would not be enabled. We use notation $p$ to index the promising placements with indicator functions enabled. Thus, Eq. A-(2) (Equation 2 in Appendix) can be written as,

$$-\frac{1}{P} \sum_{p=1}^{P} \bigtriangledown_{u_{mj}} \log \prod_{k=1}^{M} f(d_k^{(p)}|u_k^{(t)}), \tag{3}$$

where we consider only $P = \rho N$ promising placements in Eq. (9). Eq. A-(3) can be written as,

$$-\frac{1}{P} \sum_{p=1}^{P} \bigtriangledown_{u_{mj}} \sum_{k=1}^{M} \log f(d_k^{(p)}|u_k^{(t)}), \tag{4}$$

which is equivalent to,

$$-\frac{1}{P} \sum_{p=1}^{P} \bigtriangledown_{u_{mj}} \log f(d_m^{(p)}|u_m^{(t)}). \tag{5}$$

The gradient of the logarithm of a softmax distribution can be divided into two cases,

$$\bigtriangledown_{u_{mj}} \log f(d_m^{(p)}|u_m^{(t)}) = \begin{cases} 1 - f(d_m = j|u_m^{(t)}) & \text{if } d_m^{(p)} = j; \\ -f(d_m = j|u_m^{(t)}) & \text{if } d_m^{(p)} \ne j, \end{cases} \tag{6}$$

which can be reduced to a neat form,

$$\nabla_{u_{mj}} \log f(d_m^{(p)}|u_m^{(t)}) = I_{\{d_m^{(p)}=j\}} - f(d_m = j|u_m^{(t)}). \tag{7}$$

We plug Eq. A-(7) into Eq. A-(5) and let it be zero, to solve cross-entropy minimization, due to its convexity. As a result, we obtain the following probability distribution under the new parameters $u_m^{(t+1)}$,

$$f(d_m = j|u_m^{(t+1)}) = \sum_{p=1}^{P} \frac{I_{\{d_m^{(p)}=j\}}}{P}. \tag{8}$$

$\square$

## 2  Parameter Updates

In the proof, we have derived the target distribution $f(d_m|u_m^{(t+1)})$ as a function of parameters. Since we need to perform SGA steps to update parameters towards the proximal policy optimization (PPO) objective, we also need to obtain parameters $u_m^{(t+1)}$ by inverting the softmax function as follows,

$$u_{mj}^{(t+1)} = \log f(d_m = j|u_m^{(t+1)}) + C, \forall j, \tag{9}$$

where $C = \sum_{i=1}^{D} e^{u_{mi}^{(t+1)}}$ is a normalization constant. We observe that softmax distributions have the constant shift invariant property, namely, $f(d_m = j|u_m^{(t+1)}) = f(d_m = j|u_m^{(t+1)} + C), \forall C \in R$. Therefore, we can compute the optimal parameters by

$$u_{mj}^{(t+1)} = \log f(d_m = j|u_m^{(t+1)}), \forall j, \tag{10}$$

without changing the probability distribution. Our joint learning algorithm uses Eq. A-(8) and Eq. A-(10) to directly solve the global minimum of cross-entropy minimization. In general, the minimization can be efficiently solved by convex optimization techniques, such as gradient descent or Newton's method.

For each PPO update, several SGA steps are performed with respect to the PPO objective in Eq. (11). $u_m^{(t+1)}$ is first initialized as the value of $u_m^{(t)}$, and then updated by SGA steps as follows,

$$u_m^{(t+1)} = u_m^{(t)} + \alpha \frac{1}{K} \sum_{n=tK+1}^{tK+K} \frac{\nabla_{u_m^{(t+1)}} f(d_m^{(n)}|u_m^{(t+1)})}{f(d_m^{(n)}|u_m^{(t)})} (b - T(d^{(n)})) - \beta \nabla_{u_m^{(t+1)}} D_{KL}(f(d_m^{(n)}|u_m^{(t)})||f(d_m^{(n)}|u_m^{(t+1)})), \tag{11}$$

where $\alpha$ is the learning rate. For softmax distributions, the gradients in Eq. A-(11) can be derived as closed-form expressions.

## 3  Additional Evaluation

We have conducted complementary experiments to compare with a classic scheduling algorithm — Heterogeneous Earliest Finish Time (HEFT), and simple search algorithms including random search, hill climbing and genetic algorithms. To add the HEFT baseline, for each neural network, we generate a cost graph where node weight is the computation time of an operation and edge weight is the communication time (tensor size/bandwidth) between two operations. We then feed the cost graph into a HEFT scheduler to generate a schedule of operations on devices.

We compare *Post* with these additional baselines, each given 12 machines on Google Cloud with 2400 placement evaluations. For ResNet-50 on 2 GPUs (Inception-V3 on 4 GPUs), the best per-step training times (in seconds) are 1.64 (1.92) by random search, 1.52 (1.86) by hill climbing, 1.56 (2.03) by genetic algorithm, 3.41 (3.31) by HEFT and 0.86 (1.19) by *Post*. These complementary results have clearly demonstrated that more advanced optimization as *Post* is required and exhibits advantages in addressing the device placement challenge, where traditional algorithms fall short.