[Reviews · NeurIPS 2018]

Reviewer 1



This is a great work as it tackles an important problem: graph partitioning in heterogeneous/multi-device settings. There is an increasing number of problems that could benefit from resource allocation optimization techniques such as the one described in this work. ML and specifically RL techniques have been recently developed to solve the problem of device placement. This work addresses one of the main deficiencies of the prior work by making more sample efficient (as demonstrated by empirical results). The novelty is in the way the placement parameters are trained: As oppose to directly train a placement policy for best runtime, a softmax is used to model the distribution of op placements on devices (for each device among the pool of available devices.) At each iteration, the current placement distribution is refined by the conditional distribution representing the top 10% of the best placements. A KL-divergence loss is applied to minimize the distance between the old and new distribution. PPO is also used for further data sample reuse efficiency. The strengths: Great results: The results outperform existing approach based on Policy gradient by upto 58%. The overhead of placement is low given that only 2400 samples were needed to achieve the results. This makes this work pottentially deployable. Control experiments show improvement as a result of both KL minimization and PPO optimization. The results also show a clear win over prior policy gradient based work. The paper is clear and easy to follow. The code is provided and looks very modular and easy to use. Questions/suggestions: What if the sampled placements from the conditional distribution at time t+1 produces placements that are all worse than the placements produces at iteration t? Has it ever happened in your experiments? It would be good to show results on the amount of saving the optimized placements bring to the end-to-end training of the target models. Th proposed model does not seem to generalize across graphs as it uses a one hot representation for nodes. Have the authors considered ways to incorporate graph embeddings into the model? The reductions shows in Table 1 should compare Post with best baseline results, not with policy gradient.

Reviewer 2



This paper introduces a way of optimizing the device usage for accelerating neural network performance in a heterogeneous compuatational environment. It combines PPO and Cross Entopy Loss by taking a weighted average of them and optimizing the "preference" vector. The most likely device is then chosen to be the argmin of the obtained vector via the optimization process. The paper also tests the system "Post" on different heterogeneous environments with different architectures which shows that the proposed system can work with different types of architectures. However, the paper indeed has some flaws: 1. Figure 1 doesn't illustrate the author's intention properly. It's improper labelling is a hindrance to understanding what GPU ultimately gets chosen and how many GPUs are in the system. 2. In Figure 3 d) it is interesting that the post learns the pipelining operation, which are widely used in parallel computing. Since this device placement could also be viewed as scheduling problems, which could be converted as optimization problem. I wonder if author compared post with scheduling algorithms in parallel programming. 3. From the graphs provided in Figure 2 it becomes apparent that Cross Entropy beats PPO. Post combines both methods as a simple weighted average, but still beats both methods. The reason why this works is missing in the discussion. 4. In Table 1 on single GPUs Inception architectures have nearly the same performance as "Experts". There is no explanation as to why this happens.

Reviewer 3



The paper proposes combining Cross Entropy (CE) minimization with Proximal Policy Optimization (PPO) for optimizing placement decisions of operations in TensorFlow graphs across multiple devices. Results show faster and better optimization results than previous RL-based approach of Mirhoseini et al. Strengths: - The paper is written very clearly and is easy to follow. - Algorithm seems to be simpler than that proposed by Mirhoseini et al. - The results are significantly better than Mirhoseini et al. Weaknesses: - The evaluation is incomplete. The previous papers by Mirhoseini et al. unfortunately didn't compare against simple/trivial baseline algorithms like random search (try random placements and keep track of the best), hill climbing, genetic algorithms, etc. to show whether a learning approach outperforms them given the same computational budget and placement evaluation budget. This paper also doesn't address the shortcoming. So it's not at all clear whether placement optimization really requires any sophisticated optimization in the first place. Comments/Questions: - Please include in the Appendix the type of graphs shown in Figure 2 for all the entries in Table 1. This way it will be clear that the same kind of behavior seen in the three graphs in Figure 2 hold for all the settings. - Are the results in Table 1 strictly controlled for the same amount of computation and the same number of placement evaluations allowed to Policy Gradient and Post? The text does not say this. If they are not, then the results may not be meaningful. - For many TensorFlow graphs for which placement optimization is difficult, a key challenge is to stay within the memory limits of the devices. So it may be difficult to find placements, especially early on in the optimization, that satisfy memory constraints. How is this problem handled by the proposed algorithm? - TensorFlow graph running times can show high variance. What are the error bars for Figure 2 and Table 1? - Is the algorithm that is labelled as Policy Gradient a re-implementation of the Mirhoseini et al.'s approach? If so, their work relied on grouping the ops and making group-level device placements, and how were the grouping decisions made?